# Plasmodium DNA-mediated TLR9 activation of T-bet[+] B cells contributes to autoimmune anaemia during malaria

J. Rivera-Correa[1], J.J. Guthmiller[2], R. Vijay[3], C. Fernandez-Arias[1], M.A. Pardo-Ruge[1], S. Gonzalez[1], N.S. Butler[3] & A. Rodriguez[1]

Infectious pathogens contribute to the development of autoimmune disorders, but the mechanisms connecting these processes are incompletely understood. Here we show that *Plasmodium* DNA induces autoreactive responses against erythrocytes by activating a population of B cells expressing CD11c and the transcription factor T-bet, which become major producers of autoantibodies that promote malarial anaemia. Additionally, we identify parasite DNA-sensing through Toll-like receptor 9 (TLR9) along with inflammatory cytokine receptor IFN-γ receptor (IFN-γR) as essential signals that synergize to promote the development and appearance of these autoreactive T-bet[+] B cells. The lack of any of these signals ameliorates malarial anaemia during infection in a mouse model. We also identify both expansion of T-bet[+] B cells and production of anti-erythrocyte antibodies in ex vivo cultures of naive human peripheral blood mononuclear cells (PBMC) exposed to *P. falciprum* infected erythrocyte lysates. We propose that synergistic TLR9/IFN-γR activation of T-bet[+] B cells is a mechanism underlying infection-induced autoimmune-like responses.

[1] Department of Microbiology, New York University School of Medicine, New York, NY 10010, USA. [2] Department of Microbiology and Immunology, University of Oklahoma Health Sciences Center, Oklahoma City, OK 73104, USA. [3] Department of Microbiology and Immunology, University of Iowa, Iowa City, IA 52242, USA. Correspondence and requests for materials should be addressed to A.R. (email: Ana.Rodriguez@nyumc.org)

Autoimmunity during and after an infection is an extensively reported phenomenon[1, 2], but despite being frequently observed, little is known about the mechanisms underlying infection-related autoimmune responses. Malaria, a global disease caused by infection with *Plasmodium* parasites, has been associated with the development of autoimmunity in patients and mouse models[3–6]. Autoimmunity in infection has been attributed to mechanisms such as molecular mimicry but, with studies showing a broad range of self-antigen specificity in autoantibodies[5], it is generally accepted that additional mechanisms must contribute to this phenomenon.

A misguided B-cell response, that leads to production of autoreactive antibodies, is a critical pathological component of many autoimmune disorders, such as systemic lupus erythematosus[7]. The origin of these antibodies is thought to be due to the inappropriate activation of B cells, that could be induced through innate nucleic acid sensors such as Toll-like receptors (TLR)[8, 9]. TLRs, such as TLR7 and TLR9 that recognize RNA and DNA, respectively, have a critical function in innate immune sensing of various pathogens whose nucleic acids are effective pathogen-associated molecular patterns (PAMP)[10]. *Plasmodium* DNA is considered a major PAMP signal that activates TLR9 during malaria[11]. Interestingly, TLR7 and TLR9 are also important contributors of various autoimmune disorders[12, 13]. Since several infections increase the risk of developing autoimmune disorders[1, 2, 14, 15], there is a pressing need to understand the effect of autoimmunity during infection, as well as the molecular mechanisms leading to its generation.

Severe malarial anaemia is a complication reported in 50% of all severe malaria cases that is associated with mortality and morbidity. Malaria-induced anaemia is thought to be multifactorial, with the implication of bone marrow suppression, inappropriate erythropoiesis[16] and loss of infected and uninfected red blood cells (RBC)[17, 18]. Interestingly, even though *Plasmodium* parasites have an intra-erythrocytic stage, parasite driven RBC destruction contributes very little to overall anaemia since the density of infected RBC is very low. Instead, loss of uninfected RBC during malaria is considered a major contributor to anaemia in human patients and rodent models[17, 19, 20]. Various studies have described an important function of the immune system, specifically autoantibodies, in promoting anaemia during malaria[14, 21]. Particularly, antibodies against phosphatidylserine (PS) exposed on the surface of uninfected RBC during malaria promote RBC clearance by macrophages[14, 21].

Here we identify a population of autoreactive B cells that is involved in the secretion of anti-PS antibodies for anaemia induction during malaria. These atypical B cells are characterized by the expression of CD11c and T-bet, similarly to a population of autoreactive B cells described in ageing-related and chronic autoimmune disorders[22, 23]. These malaria-induced autoreactive T-bet[+] B cells are generated through the activation of three receptors that act synergistically: interferon-γ receptor (IFN-γR), the B-cell receptor (BCR) and TLR9 that senses *Plasmodium* DNA. Our results suggest that autoreactive B cells are activated by pathogen PAMPs during infection, linking autoimmunity and infectious diseases.

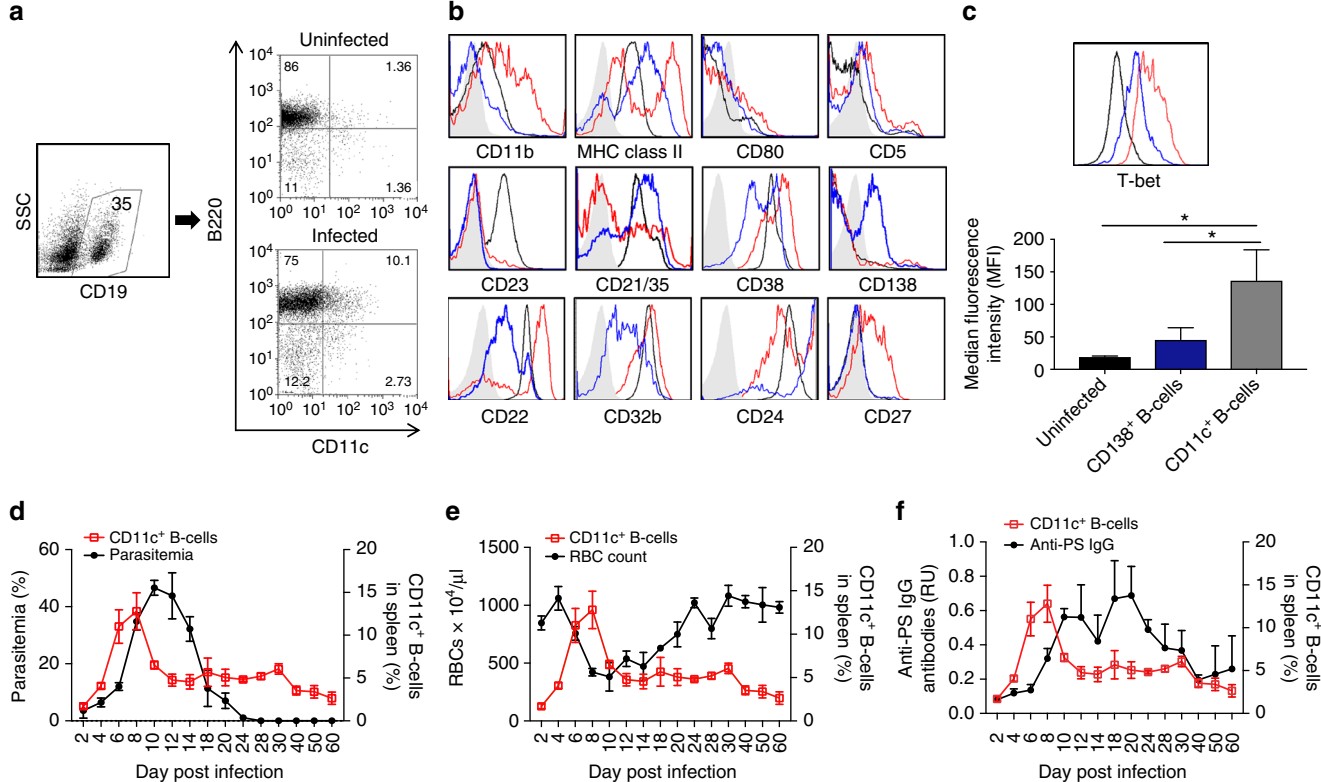

**Fig. 1** CD11c[+] T-bet[+] B cells expand during *Plasmodium* infection and correlate with severe anaemia. Gating scheme **a** and representative plots **b** for surface marker characterization in gated CD19[+] splenocytes from uninfected (black line, B220[high]) or *P. yoelii*-infected mice at day 10 post infection gated for B220[high] CD11c[+] (red line) or for B220[low]CD138[+] (blue line, conventional plasmablasts and plasma cells), with respective isotype control antibodies (gray shadow). **c** Representative plots and quantification of T-bet-expressing B cells within the gates described in (**b**). Experiments in **a**, **b** were repeated 2 times. **d**, **f** Quantification of CD11c[+] B cells in spleen (gated as described in **b**, red line) across the days of infection, compared to parasitemia (**d**), anaemia development (**e**) and anti-PS IgG antibodies (RU relative units) (**f**) (black line). Each data point represents the mean ± SD of n = 3 mice. Different groups of three mice were killed and analyzed on each day of infection. Significance determined by unpaired Student's t test *p < 0.05

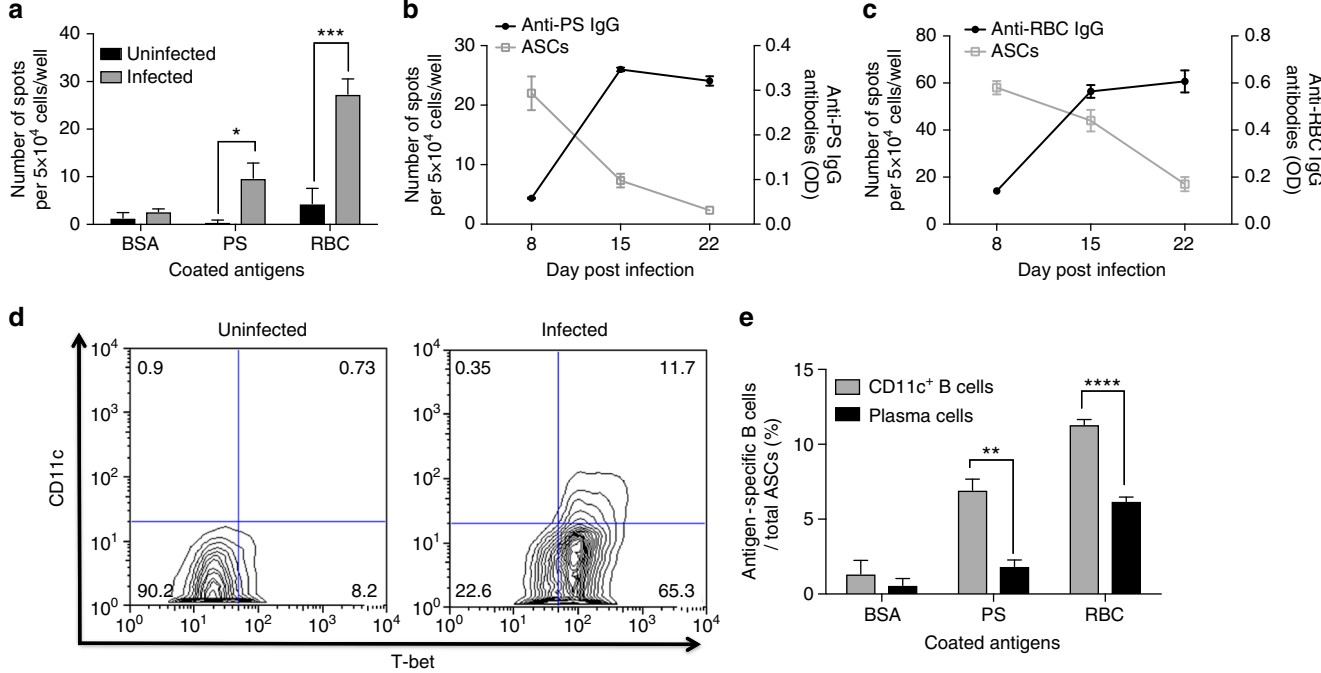

**Fig. 2** CD11c[+] T-bet[+] B cells are the main producers of autoantibodies during *P. yoelii* infection. **a** B-cell ELISPOT of splenocytes from uninfected and *P. yoelii*-infected mice at day 10 post infection. **b**, **c** Quantification of antibody-secreting cells (gray squares) and antibodies (black circles) against PS (**b**) and RBC (**c**) antigens and across *P. yoelii* infection. **d** Representative plots of gated CD19[+] splenocytes to identify CD11c[+] T-bet[+] B cells. **e** B-cell ELISPOT on enriched CD11c[+] B cells and CD138[+] plasma cells from *P. yoelii*-infected mice at day 6 post infection. Each data point represent the mean ± SD of *n* = 3 mice. Experiments were repeated 2 times, one representative example is shown. Significance determined by unpaired Student's *t* test *$p < 0.05$, **$p < 0.01$, ***$p < 0.001$, ****$p < 0.0001$

## Results

**T-bet[+] CD11c[+] B cells during development of malarial anaemia.** Malaria induces the development of an autoimmune antibody response both in humans and murine models[5]. We have described the development of anti-PS (anti-PS) antibodies and their direct role in promoting malarial anaemia during *P. yoelii*-XNL17 infection in mice and the correlation of anaemia and anti-PS antibodies in *P. falciparum* patients with post-malarial anaemia[14]. Using *P. yoelii* infection of Swiss Webster mice as a model, we aimed to identify the splenic B-cell population producing anti-PS antibodies during malarial anaemia. After gating out non-B cells (CD19−) in splenic lymphocytes, we identified an atypical population of B cells, as defined by high expression of CD11c and B220, which expanded in infected mice (Fig. 1a). We further characterized these cells and identified them as a distinct population from classically activated plasmablasts or plasma cells (CD138[+]B220[low]). CD11c[+] B220[+] cells also express a more innate-like phenotype (CD11b[+], MHC-II[+]) distinct from other B-cell subpopulations and presents a highly activated phenotype as defined by surface marker expression such as CD27. We also observed partial downregulation of inhibitory receptors, such as CD22 and CD32b, and a threshold regulator of B-cell activation, CD23, suggesting that this population may be prone to spontaneous autoreactivity (Fig. 1b).

A similar population denominated age-associated B cells (ABC) has been described to accumulate during age and in various murine models of autoimmune disorders[9, 23, 24]. Based on the strong phenotype similarity of the ABCs and the CD11c[+] B-cell population in *P. yoelli*-infected mice, we assessed the expression of the transcription factor T-bet, which is expressed by the ABC population. Intracellular staining revealed that virtually all CD11c[+] B cells in infected mice express the transcription factor T-bet (>99%), which is not significantly expressed

($p = 0.06$, Student's *t* test) in classically activated plasma cells (Fig. 1c). The generalized T-bet expression in CD11c[+] B cells is also similar to the autoreactive ABC population where CD11c expression is secondary to T-bet expression on B cells[25].

Analysis of dynamic parameters throughout infection showed that the expansion of CD11c[+]T-bet[+] B cells correlates directly with increasing parasitemia (Fig. 1d), and inversely with RBC density (Fig. 1e), which is consistent with CD11c[+] T-bet[+] B cells having a role in anaemia. We also observed that circulating anti-PS IgG antibodies appear shortly after the expansion of CD11c[+] T-bet[+] B cells and they both decrease at similar rates, persisting at lower levels after parasites have been cleared (Fig. 1f).

**CD11c[+] T-bet[+] B cells secret anti-RBC antibodies in malaria.** We determined the capacity of B cells to produce autoreactive antibodies during *P. yoelii* infection by B-cell ELISPOT. Plates were coated with PS or lysates of uninfected RBCs and incubated with splenocytes from infected (day 10) and control uninfected mice. We observed the appearance of spots in wells incubated with splenocytes from infected mice, each spot representing one antibody-secreting cell recognizing the coated antigen (Fig. 2a). We also assessed the dynamics of antibody-secreting cells against PS and whole-uninfected RBC lysates finding that specific antibody-secreting cell responses were already high at day 8 after infection followed by antibody levels that rise later. Also, antibody-secreting cells against PS and RBC antigens decrease with parasitemia while PS-specific and RBC-specific antibodies levels are still high (Fig. 2b, c).

We next compared the capacity of conventional plasma cells (CD138[+]) and CD11c[+] T-bet[+] B cells to produce autoreactive antibodies during *P. yoelii* infection. To enrich for the atypical CD11c[+] T-bet[+] B-cell population, we took advantage of their

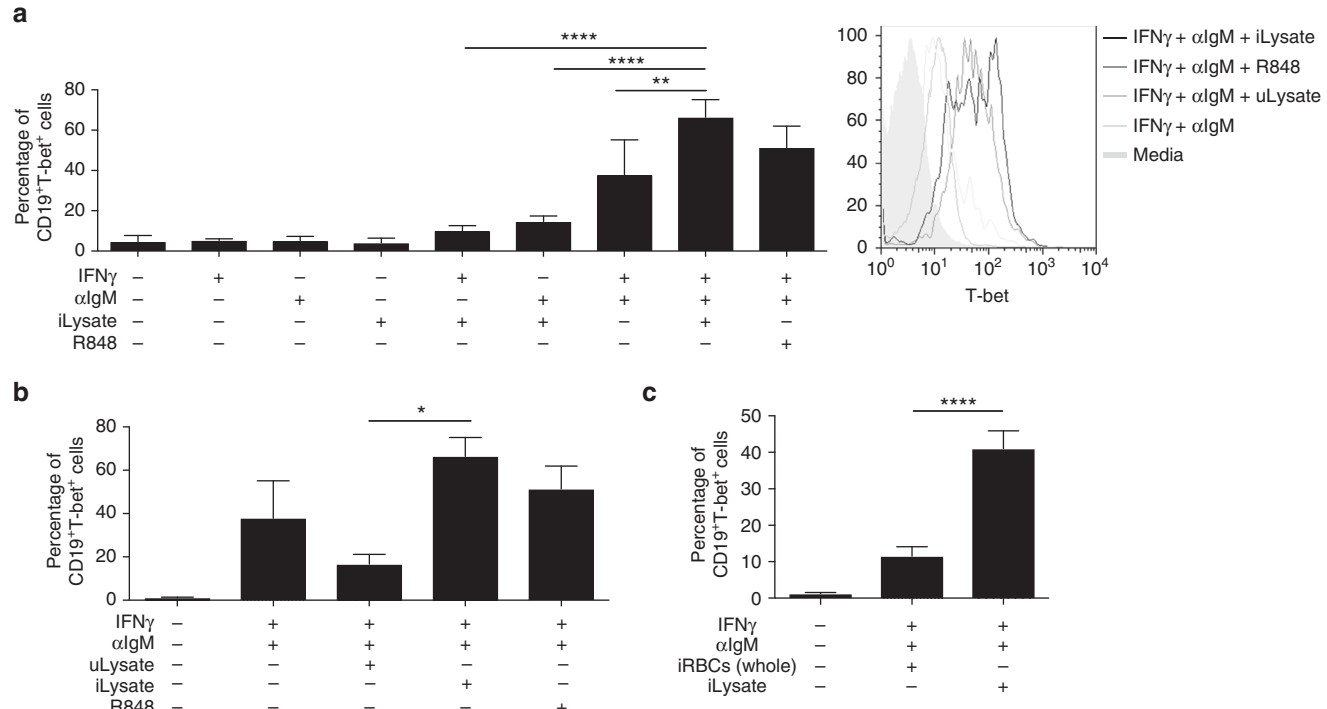

**Fig. 3** IFN-γ synergizes with *Plasmodium* molecules to induce T-bet expression on B cells in vitro. **a–c** Purified naive B cells (CD43⁻) from uninfected mice were cultured under the indicated conditions for 3 days, when T-bet expression was determined in CD19⁺ cells. *P. yoelii*-infected RBC lysate (iLysate), uninfected RBC lysate (uLysate), agonist of TLR7 (R848). Representative histogram of T-bet expression (right panel, a) of the indicated conditions. Experiments were repeated 2 times, one representative example is shown. Bars represent the means ± SD of $n = 3$ mice. Significance determined by one-way ANOVA *$p < 0.05$, **$p < 0.01$, ***$p < 0.001$, ****$p < 0.0001$

surface marker CD11c, since virtually all CD11c⁺ B cells in the spleen of *P. yoelii*-infected mice are T-bet⁺ (Fig. 2d). Since this surface marker is not expressed in conventional antibody-secreting plasma cells, these will not be enriched under CD11c selection. This method allowed for efficient enrichment of CD11c⁺ B cells (CD19⁺) along with CD11c⁺ non-B cells (CD19⁻) (Supplementary Fig. 1a) that should include mostly dendritic cells[26] which are inert in the B-cell ELISPOT since they do not produce antibodies. In parallel, classical plasma cells were enriched using their surface marker CD138 (Supplementary Fig. 1b), which is not expressed in the surface of the CD11c⁺ T-bet⁺ B-cell population (Fig. 1a). After the enrichment of both classical (CD138⁺) and atypical (CD11c⁺T-bet⁺) populations, we assessed their reactivity against PS and whole-uninfected RBC lysate antigens finding that atypical B cells were proportionally more efficient antibody-secreting cells against PS and whole RBC lysates compared to classical plasma cells (Fig. 2e). Antibodies and antibody-secreting cells recognizing other phospholipids and DNA are also found in infected mice (Supplementary Fig. 2). These results indicate that CD11c⁺ T-bet⁺ B cells produce antibodies against PS and other self-antigens, probably contributing to the anti-PS response that promotes malarial anaemia.

**Infected RBCs and IFN-γ induce T-bet in B cells**. To understand the mechanism by which autoreactive CD11c⁺ T-bet⁺ B cells expand during malaria we first determined the effect that *Plasmodium*-infected RBCs can directly have on promoting B-cell autoreactivity. It is well characterized that intracellular expression of T-bet, which has been reported in autoreactive B cells[27], is highly dependent of IFN-γ signalling[28], establishing it as a necessary signal for the expression of this transcription factor. Since IFN-γ is produced in high concentrations in the malaria

immune response[29], it is likely that B cells will be exposed to this cytokine during infection.

To determine whether *Plasmodium*-infected RBCs can induce the expression of T-bet on B cells, we isolated naive B cells from control uninfected mice and incubated them in vitro with a lysate of *Plasmodium*-infected RBCs, either alone or in combination with IFN-γ. BCR crosslinking (anti-IgM) was added to bypass the need for antigen specificity. We observed that addition of each of the three signals (IFN-γ, anti-IgM, and a *P. yoelii*-infected RBC lysate) independently induces a very minor increase in T-bet expression in B cells, but combinations of these signals two-by-two, and especially when the three were added together, results in a synergistic increase in T-bet⁺ B cells (Fig. 3a). It should be noted that this high level of T-bet expression on B cells is similar to the level obtained when an agonist of TLR7 together with IFN-γ and anti-IgM were added, as previously described[9, 25]. To confirm the specificity of the response, we compared the lysate of uninfected and infected RBCs observing that only the latter induces activation of T-bet expression (Fig. 3b). We also observed that whole intact infected RBCs are not able to induce T-bet expression as compared to infected RBC lysates (Fig. 3c), suggesting that the signaling factor may be released upon rupture of infected RBCs. Taken together, these data attribute a direct role for *Plasmodium*-infected RBCs in promoting B-cell autoreactivity.

***Plasmodium* DNA triggers B-cell T-bet expression through TLR9**. To further dissect the role of *Plasmodium*-infected RBC lysates on promoting B-cell autoreactivity, we set out to identify the *Plasmodium*-specific signal and the receptor that is sensing it. In various autoimmune models it has been proposed that dual-engagement of the BCR along with nucleic acid sensor TLRs can promote T-bet expression along with autoantibody

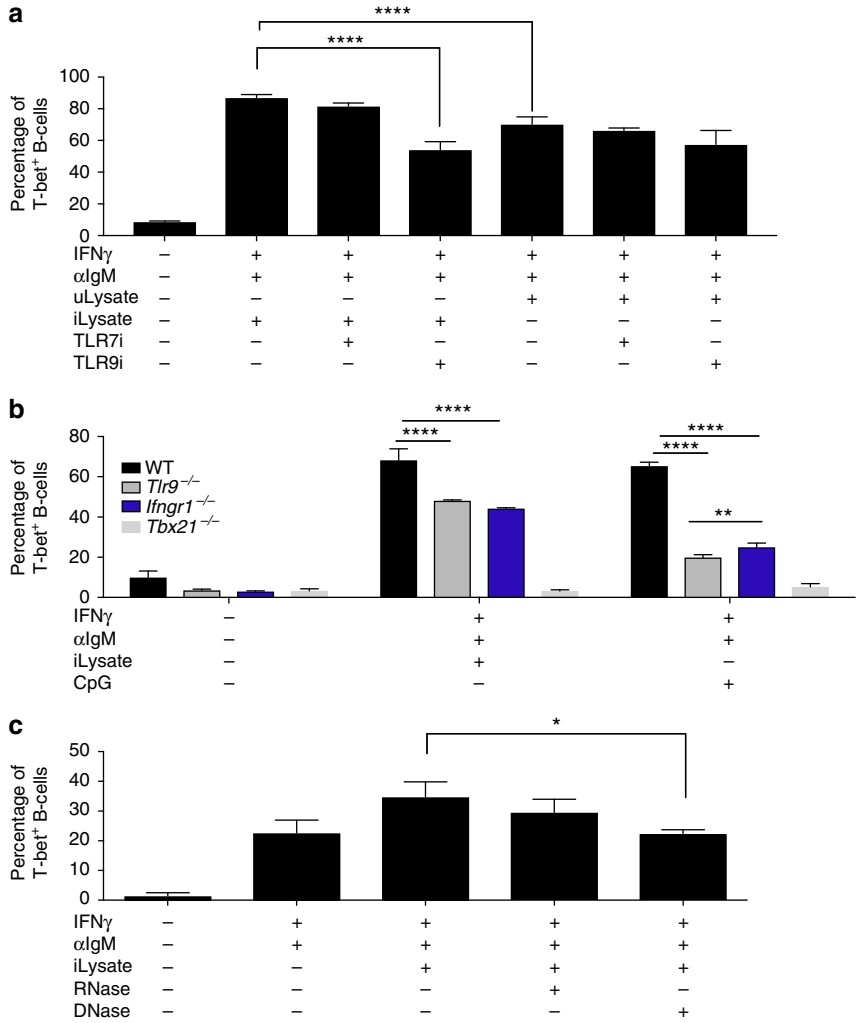

**Fig. 4** *Plasmodium* DNA enhances IFN-γ-dependent T-bet expression through B-cell intrinsic TLR9 in vitro. Purified naive B cells (CD43⁻) from uninfected mice were cultured under the indicated conditions for 3 days, when T-bet expression was determined in CD19⁺ cells from control uninfected WT mice (**a**, **c**) or from *Ifngr1⁻/⁻*, *Tbx21⁻/⁻* or *Tlr9⁻/⁻* mice. **b** *P. yoelii*-infected RBC lysate (iLysate), uninfected RBC lysate (uLysate), antagonists of TLR7 or TLR9 (ODN 20958 TLR7i or A151 TLR9i), DNase, or RNase treated iLysates. Experiments were repeated two times, one representative example is shown. Bars represent the means ± SD of $n = 3$ mice (**a**, **c**) or of triplicated samples from one mouse (**b**). Significance determined by one-way ANOVA *$p < 0.05$, **$p < 0.01$, ****$p < 0.0001$

production[8, 9]. To test whether *Plasmodium* could be enhancing T-bet expression through B-cell intrinsic nucleic acid sensor TLRs, we performed the in vitro activation of naive B cells as shown above (Fig. 3) in the presence of antagonists for TLR7 and TLR9. Remarkably, the *Plasmodium*-dependent enhancement of intracellular T-bet expression was reduced to the level of the two general signals (IFN-γ⁺ anti-IgM) in the presence of the antagonist for TLR9 (ODN A151) but not in the presence of the antagonist for TLR7 (ODN 20958). This effect is *Plasmodium*-specific since neither of the antagonists had an inhibitory effect when uninfected RBC lysates were added (Fig. 4a). Addition of the antagonists for TLR7 and 9 together did not result in a significant ($p = 0.89$, one-way ANOVA) difference compared to the antagonist for TLR9 alone, suggesting that the role of TLR7 is minor. An antagonist for TLR7, 8, and 9 (ODN 2088) also did not increase considerably the inhibitory effect (Supplementary Fig. 3a).

Additionally, these data were validated with a genetic model by performing the activation assay of naive B cells from *Tlr9⁻/⁻* uninfected mice, which have a lower level of T-bet expression when compared to WT control B cells (Fig. 4b). Similar results were found using naive B cells from mice deficient in MyD88, an adaptor molecule required for TLR9 signalling[30] (Supplementary Fig. 3b).

Activation of naive B cells from *Ifngr1⁻/⁻* also results in decreased T-bet expression, consistent with our previous results indicating that this process is dependent on IFN-γ. Naive B cells from T-bet-deficient mice (*Tbx21⁻/⁻*) were used as negative control (Fig. 4b).

The decrease in T-bet expression caused by inhibition of TLR9 signaling suggests that DNA from *Plasmodium* is likely the activating signal that infected RBC lysates provide to B cells. Indeed, DNase treatment, but not RNase treatment, of *Plasmodium*-infected RBC lysates eliminates their enhancing effect on intracellular T-bet expression on B cells (Fig. 4c). Taken together, these data reveal *Plasmodium* DNA as the activating signal and B-cell intrinsic TLR9 as the receptor that enhances intracellular T-bet expression.

**Deficiency in IFN-γR T-bet or TLR9 reduces malarial anaemia.** To determine the in vivo relevance of T-bet⁺ B cells in malarial

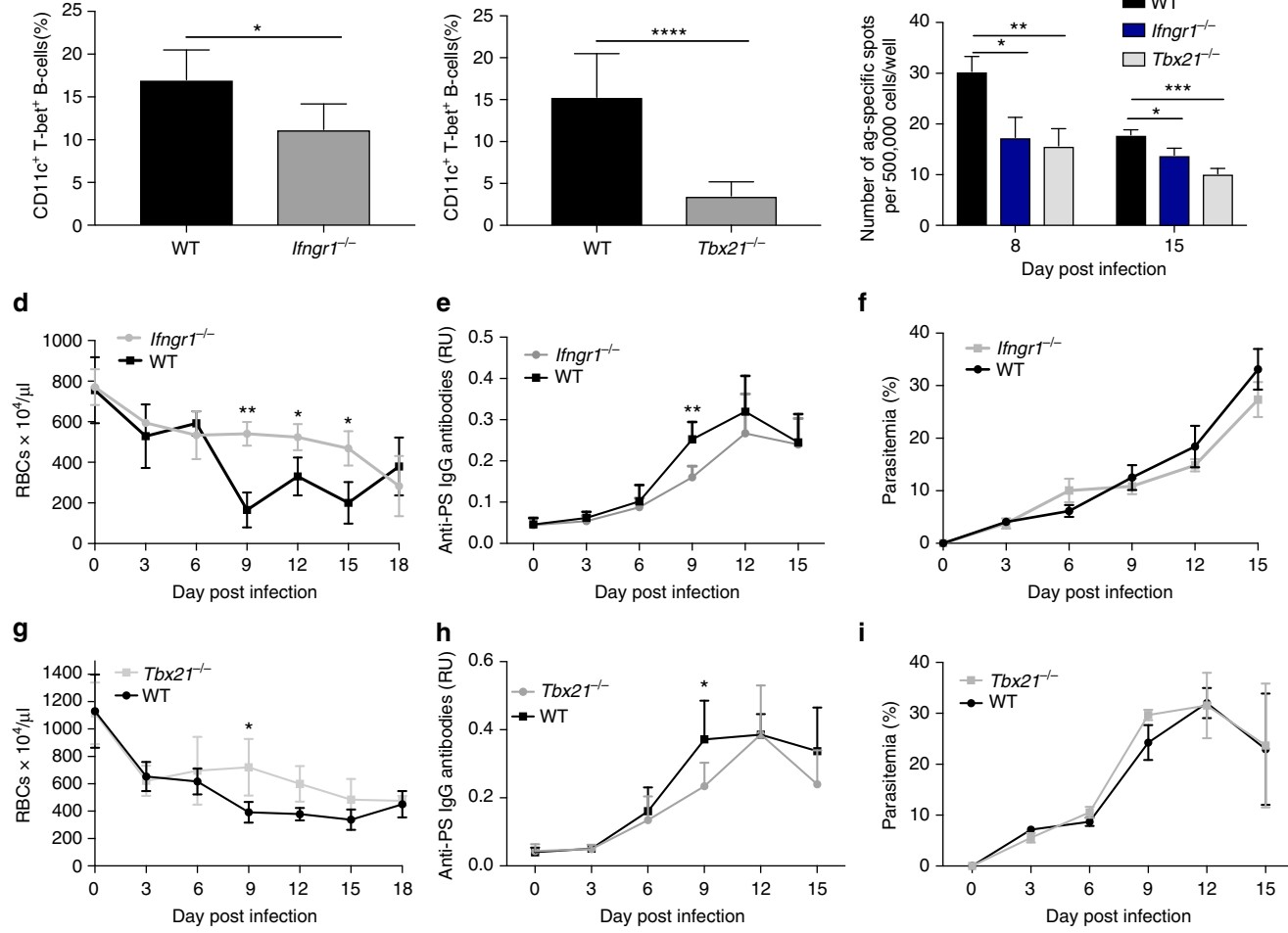

**Fig. 5** Deficiency in IFN-γ or T-bet confers partial protection against malarial anaemia in *P. yoelii*-infected mice. **a, b** Quantification of CD11c+ T-bet+ cells from gated CD19+ splenocytes from healthy WT naive mice (black bars), *Ifngr1*−/− (**a**, gray bar), or *Tbx21*−/− (**b**, gray bar). **c** PS B-cell ELISPOT of splenocytes from *P. yoelii*-infected WT, *Ifngr1*−/− or *Tbx21*−/− from mice at day 8 and 15 post infection. **d, i** Quantification of anaemia development (**d, g**), anti-PS IgG antibodies (**e, h**) and parasitemia (**f, i**) across the days of infection of *P. yoelii*-infected WT (black line), *Ifngr1*−/− (d-f, gray line) or *Tbx21*−/− (**g–i**, gray line) mice. RU relative units. Bars and line graphs represent the mean ± SD of *n* = 3 mice. Experiments were repeated 2 times, one representative example is shown. Significance determined by unpaired Student's *t* test *p < 0.05, **p < 0.01, ****p < 0.0001

anaemia, we infected *Ifngr1*−/−, *Tbx21*−/− and *Tlr9*−/− mice with *P. yoelii* and followed the levels of splenic CD11c+ T-bet+ B cells, anti-PS IgG antibodies and the development of anaemia. Mice deficient in IFN-γR demonstrated a reduction in CD11c+ T-bet+ B cells compared to their respective WT-infected controls (Fig. 5a), indicating that IFN-γ signaling contributes to the generation of these cells. T-bet-deficient mice (*Tbx21*−/−) also have a reduction in CD11c+ T-bet+ B cells (Fig. 5b). Since these mice do not express T-bet, analysis of T-bet expression serves as negative control for intracellular T-bet staining (Supplementary Fig. 4a). A reduction in the antibody-secreting cell response to PS was also found in both *Ifngr1*−/− and *Tbx21*−/− mice (Fig. 5c).

To confirm the identity of the population in these mice, we assessed the presence of CD11c+ B220+ B cells on *Tbx21*−/− mice as described in Fig. 1a, which have a similar reduction (Supplementary Fig. 4b). Both *Ifngr1*−/− and *Tbx21*−/− mice had partial protection against malarial anaemia (Fig. 5d, g) that correlated with a reduction of circulating anti-PS IgG antibodies compared to their WT counterparts (Fig. 5e, h). Parasitemia was similar for both deficient and WT mice (Fig. 5f, i) during the time period analyzed, excluding the possibility of any effects mediated by different levels of infection. Collectively, these data suggest a role for both IFN-γR and T-bet, in promoting the expansion of

CD11c+T-bet+ B cells, the generation of anti-PS antibodies and malarial anaemia during *P. yoelii* infection. Similarly to the in vitro data (Fig. 3), the removal of IFN-γ, one of the three signals contributing to the expression of T-bet in B cells, has a detectable but moderate effect in the levels of these cells. This decrease is presumably the cause of the lower levels of anti-PS antibodies, which in turn results in decreased anaemia in the mice.

To determine whether the effects of IFN-γR and T-bet are intrinsic to the B-cell population, we analyzed the levels of anti-PS antibodies in mixed bone marrow chimeric mice in which either IFN-γR or T-bet deficiency is restricted to the B-cell compartment[31]. We found that the levels of anti-PS antibodies generated during *P. yoelii* infection were lower in these chimeric mice compared to control chimeric mice harboring WT B cells (Fig. 6), confirming a direct role of B-cell IFN-γR and T-bet in the generation of anti-PS antibodies, presumably through the expansion of the CD11c+T-bet+ B-cell population. We observed higher levels of anti-parasite (merozoite surface protein 1, MSP1) antibodies (Fig. 6c), as expected, since total anti-parasite-specific antibodies were found to be elevated in these mice[31]. These results confirm the specific dependency of anti-PS antibodies on B-cell intrinsic IFN-γR and T-bet.

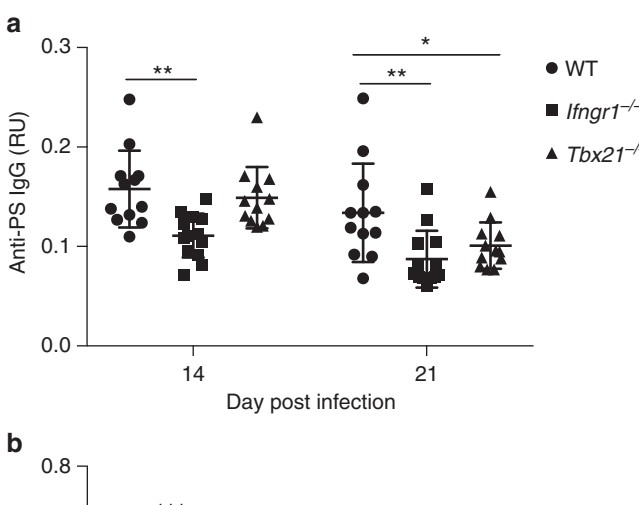

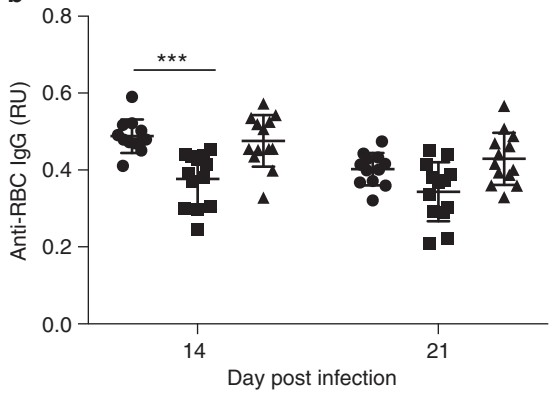

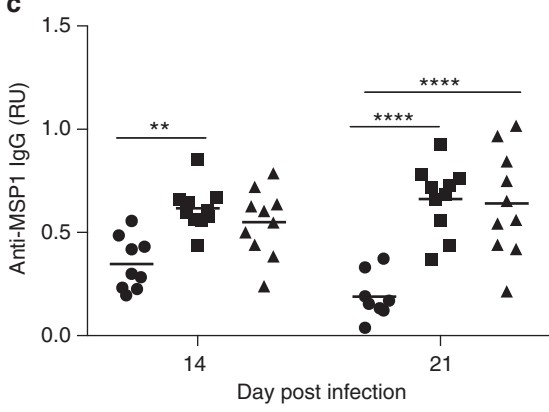

**Fig. 6** B-cell-specific deletion of IFN-γR or T-bet leads to a reduced anti-PS B-cell response during *P. yoelii* infection. **a–c** Quantification of anti-PS, anti-RBC lysate or anti-MSP-1 IgG antibodies (RU relative units) from mixed bone marrow chimeras harboring either WT (circle), *Ifngr1*[−/−] (square), *Tbx21*[−/−] (triangle) B cells (**a**, **b**). Significance determined by two-way ANOVA followed by F-statistic test. Graphs represent the individual values of $n = 11$ mice with the mean $*p < 0.05$, $**p < 0.01$, $***p < 0.001$, $****p < 0.0001$

Another contributing signal for the expansion of T-bet[+] B cells found in vitro was the innate immune DNA sensor TLR9 (Fig. 4). Consistently, *P. yoelii*-infected mice deficient for TLR9 had a reduction in CD11c[+] T-bet[+] B cells (Fig. 7a) and partial protection against malarial anaemia (Fig. 7b) that correlated with a mild but significant reduction ($p < 0.05$, Student's *t* test) in circulating anti-PS IgG antibodies (Fig. 7c) compared to WT controls at days of similar parasitemia (Fig. 7d). A similar reduction in the antibody-secreting cell response to PS was also found (Fig. 7e), confirming a role for this DNA sensor in the anti-

PS response at the cellular level. Interestingly, TLR9-deficient mice are known to develop general autoimmunity due to spontaneous B-cell activation[32]. Indeed, this was evidenced in *P. yoelii*-infected *Tlr9*[−/−] mice by their higher anti-dsDNA titers compared to their WT controls (Supplementary Fig. 5).

To determine whether the effects of TLR9 are intrinsic to the B-cell population, we analyzed the levels of anti-PS antibodies in mixed bone marrow chimeric mice in which TLR9 deficiency is restricted to the B-cell compartment. We infected control and chimeric mice with *P. yoelii* and measured PS-specific and parasite-specific (MSP1) secreted antibody responses using ELISA and ELISPOT. Although the levels of anti-PS antibodies were not significantly different in the *Tlr9*[−/−] B-cell chimeric mice when compared with controls, we observed a significant reduction in the levels of anti-PS secreting B cells by ELISPOT (Supplementary Fig. 6). Large differences in B-cell numbers between WT and *Tlr9*[−/−] B-cell chimeric mice, suggest a role for TLR9 in B-cell development and possibly explain the lack of significant differences in anti-PS responses by ELISA in contrast with a more sensitive assay such as the ELISPOT.

Collectively, these results suggest a role for TLR9, IFN-γR and T-bet as regulators of anti-erythrocyte autoimmunity and promoters of malarial anaemia during *P. yoelii* infection, possibly mediated by CD11c[+] T-bet[+] B cells.

**P. falciparum stimulates T-bet[+] B cells and autoantibodies**. To test whether a similar mechanism of activation of B cells would also be induced by the human parasite *P. falciparum*, we used in vitro cultures of peripheral blood mononuclear cells (PBMCs) from healthy naive donors incubated with *P. falciparum*-infected RBC lysates. Analysis of CD19[+] cells, revealed a significant expansion ($p < 0.05$, one-way ANOVA) of T-bet[+] B cells in these cultures, that was not observed when PBMCs were incubated with uninfected RBC lysates (Fig. 8a, b).

We next asked whether exposure to *P. falciparum*-infected RBC lysates alone could lead to autoantibody production in vitro. We hypothesized that *P. falciparum*-infected RBC lysates would be able provide all the signals required for the expansion of T-bet[+] B cells and production of anti-PS antibodies, since the activation of TLR9, BCR and IFN-γR could be induced respectively by *P. falciparum* DNA, the antigen PS (both present in the infected RBC lysate) and IFN-γ, that could be induced by the parasite in PBMCs[33]. Indeed, exposing naive PBMCs to *P. falciparum*-infected RBC lysates led to potent IFN-γ production (Fig. 8c), confirming that this cytokine is present in the co-culture.

Strikingly, exposure of PBMCs to *P. falciparum*-infected RBC lysates was sufficient to induce potent anti-PS and anti-RBC antibody production (Fig. 8d, e) respectively. Taken together, these data suggest that an expansion of T-bet[+] B cells could mediate autoreactivity during human malaria *P. falciparum* infection.

## Discussion

Autoimmune antibodies against various antigens have been described during malaria[5] but also during several acute infections characterized by a highly inflammatory response[2]. Our findings suggest that the generation of CD11c[+] T-bet[+] B cells as major mediators of the autoreactive antibody response to RBC and, in particular to PS, may be analogous to the generation of specific populations of autoreactive B cells in chronic autoimmune disorders, where they are viewed as a "dichotomous" immune response[24]. Autoimmunity during infection may have developed as an additional pathway to fight pathogens by targeting infected host cells. In the case of anti-PS antibodies, recognition of stressed

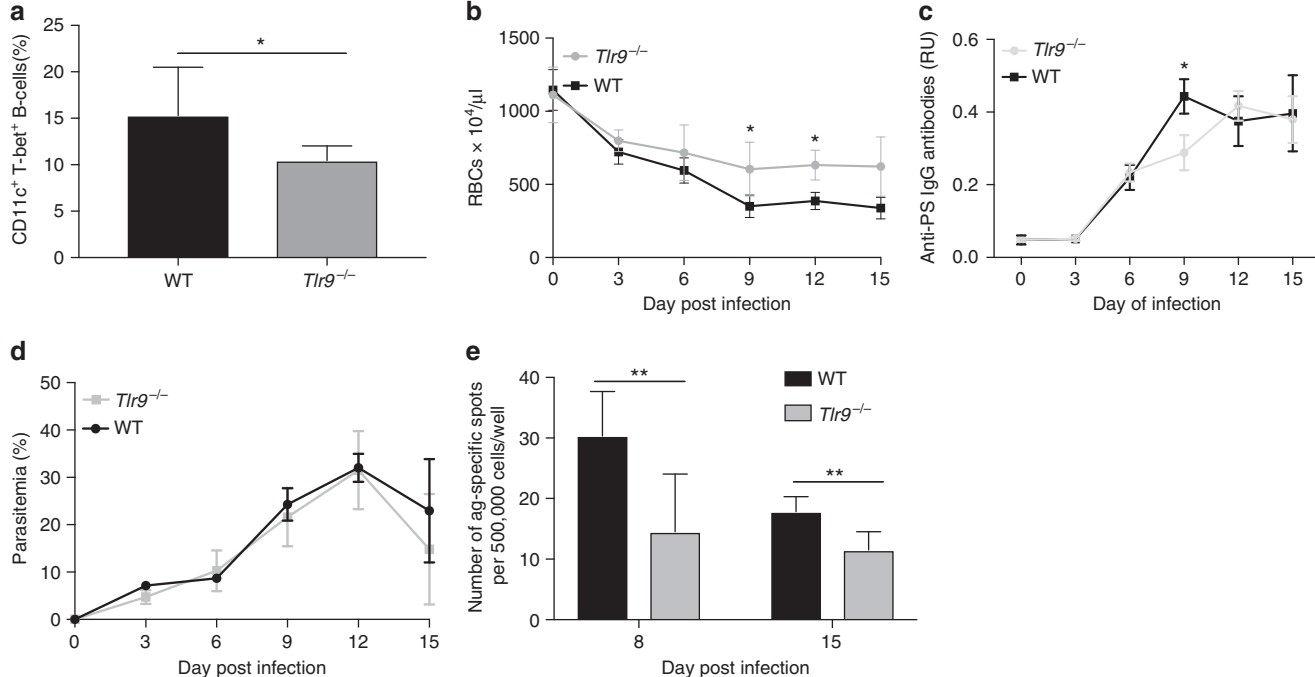

**Fig. 7** TLR9 regulates malarial anaemia and autoimmune responses in *P. yoelii*-infected mice. **a–d** Quantification of CD11c[+] T-bet[+] B cells (**a**), anaemia development (**b**), anti-PS IgG antibodies (RU relative units) (**c**), and parasitemia (**d**) of *P. yoelii*-infected WT (black line or bar) or *Tlr9[−/−]* (gray line or bar) mice at different days after infection. **e** PS B-cell ELISPOT of splenocytes from *P. yoelii*-infected WT (black bars) or from *Tlr9[−/−]* (gray bars) mice at days 8 and 15 post infection. Experiments were repeated 2 times, one representative example is shown. Bars and line graphs represent the means ± SD of *n* = 3 mice (**a–d**) or *n* = 2 (**e**). Significance determined by unpaired Student's *t* test *$p < 0.05$

cells that are exposing PS may serve as a mechanism to target infected cells. In this context, autoimmunity would not be seen as a "misguided" immune response, but as an additional immunological tool to fight infections, which in the case of malaria also results in the elimination of uninfected RBCs that expose PS, contributing to anaemia[14].

This study is focused in understanding the basic mechanisms underlying the development of autoimmune-like responses during *Plasmodium* infection. We have identified an atypical population of CD11c[+] T-bet[+] B cells as major mediators of the autoreactive antibody response to PS, contributing to malarial anaemia. Additionally, our results highlight a novel role for a pathogen PAMP (parasite DNA) as a direct activator of B cells through TLR9, promoting the expression of the transcription factor T-bet and the secretion of autoantibodies in synergy with the ligation of the BCR and IFN-γR. In the context of malaria, it is likely that the signals required for the activation of these autoreactive B cells could converge in vivo since *Plasmodium* DNA[34], as well as host membrane components, such as antigenic microparticles that expose PS[35] or even PS derived from *Plasmodium*[36], and IFN-γ[29] are found in the circulation of patients during acute infection.

TLR9 has a prominent role in the innate immunity to malaria, where it is thought to be a sensor of parasite DNA contributing decisively to malaria-induced inflammation[11] IFN-γ has a major role in controlling *Plasmodium* infection, but it can also exacerbate the disease and dampen protective immunity[31, 37]

Our experiments with the transgenic mouse lines, *Ifngr1[−/−]* and *Tbx21[−/−]*, had partial protection against malarial anaemia correlating with decreased anti-PS titers and CD11c[+] T-bet[+] B cells, demonstrating the relevance that these molecules have in the pathology and indicating that T-bet is not only a marker, but also has an important role in the anti-PS response, which is consistent with previous findings[28]. We also observed a decrease

in anti-PS antibodies in chimeric mice with B cells lacking either IFN-γR or T-bet. Importantly, parasite (MSP1)-specific antibody responses are elevated in chimeric mice with B cells lacking either IFN-γR or T-bet, suggesting that the lack of expression of IFN-γR or T-bet in B cells selectively affects the production of anti-PS antibodies but not the quantity or quality of germinal center-derived parasite-specific antibodies[31].

These results follow the lines of several non-infectious autoimmune models where ABCs that express T-bet appear during aging of mice[23] and in various autoimmune models[22, 25, 38]. In contrast to the T-bet[+] B-cell population in malaria and in B cells that accumulate in aged mice[23] that are activated by TLR9, the ssRNA sensor TLR7 is thought to be the major promoter driving the differentiation of the T-bet[+] B-cell population in mice models of autoimmune disorders[9]. Although in B cells both TLR7 and TLR9 have a similar effect inducing the expression of T-bet and promoting autoreactivity[24], these receptors have different roles as systemic drivers of autoimmunity in different mice models. While overexpression of TLR7 is a major initiator of autoimmunity[13], a deficiency in TLR9 can either exacerbate or ameliorate autoimmunity[32]. Our studies in *Tlr9[−/−]* mice need to be interpreted in this context of specific anti-PS autoreactivity[39], where we observed reduced levels of splenic CD11c[+] T-bet[+] B cells, correlating with lower levels of anti-PS antibodies and anti-PS-specific antibody-secreting cells despite increased anti-dsDNA titers. This dichotomy in anti-PS vs. anti-DNA autoantibody levels in *Tlr9[−/−]* infected mice suggests that CD11c[+] T-bet[+] B cells may represent a heterogeneous population whose autoantibody specificity depends on the signals received for activation and on the environment determined by specific infections or pathological settings.

This study provides an alternative mechanism for autoimmune-like responses during an infection and a novel role for a pathogen in directly promoting autoreactivity.

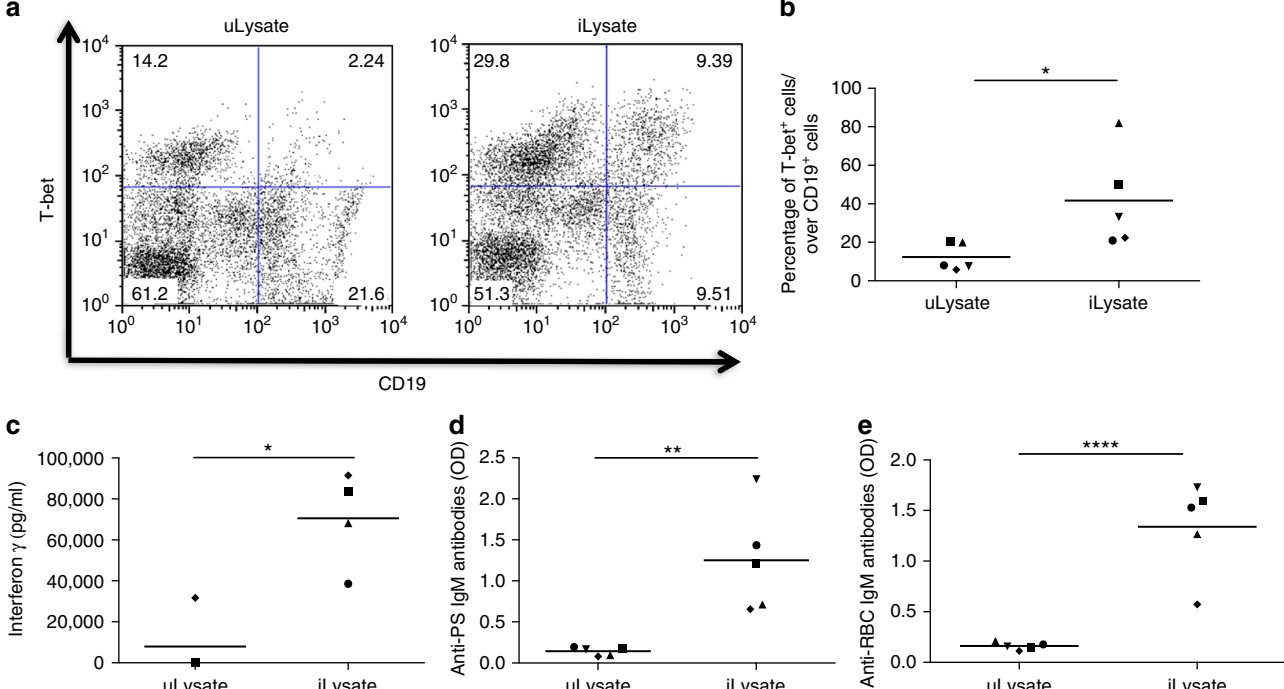

**Fig. 8** *P. falciparum* drives expansion of human T-bet[+] B cells and production of anti-PS and anti-RBC antibodies in vitro. Representative plots **a** and quantification **b** of CD19[+] T-bet[+] cells from human naive PBMCs cultured for 6 days with uninfected RBC lysates (uLysate) or *P. falciparum*-infected RBC lysates (iLysate). Detection of IFN-γ (**c**), anti-PS IgM (**d**), and anti-RBC IgM (**e**) antibodies in the supernatants of PBMCs cultured for 11 days. Each symbol represents a result from one individual donor and experiment. Significance determined by matched one-way ANOVA. Graphs represent the grand means of $n = 5$ donors *$p < 0.05$, **$p < 0.01$, ****$p < 0.0001$

Autoimmune-like responses during infection have been understudied, and generally attributed to immunological phenomena such as molecular mimicry and bystander activation[2]. Our results in mice demonstrate the synergistic effect that TLR9 activation by *Plasmodium* DNA has with BCR engagement and IFN-γ directly on promoting B-cell autoreactivity, through robust T-bet induction. T-bet expression on B cells, as well as IFN-γ stimulation, have been previously linked with production of autoreactive IgG antibodies and autoimmunity[28], emphasizing the presence of shared triggers between malaria and autoimmune disorders[27, 40].

The generation of T-bet[+] B cells and anti-PS antibodies by human naive PBMCs exposed to *P. falciparum*-infected RBC lysates indicates that this mechanism of atypical B-cell activation may be similar in humans. Exposure of naive PBMCs to infected lysates alone provided a robust IFN-γ, and remarkably, it also led to the expansion of T-bet[+] B cells and the generation of anti-PS and anti-RBC. These results provide a plausible mechanism for the generation of autoreactive antibodies that during decades has been described in malaria in both mice models and patients[4–6, 41, 42]. Additionally, we propose that this mechanism could be shared with other infectious diseases that lead to autoimmune-like responses where the pathogen nucleic acids are sensed by TLRs. Having identified parasitic DNA and IFN-γ as a key factors in the generation of autoreactive antibodies, suggests that other highly inflammatory lytic infections that result in elevated DNA levels in the circulation may lead to B-cell autoreactivity, similarly as it has been proposed in chronic autoimmune disorders such as systemic lupus erythematous[43]. In conclusion, these studies demonstrate an alternative mechanism by which a pathogen can directly lead to the development of an autoreactive response and contribute to pathology during an infection.

## Methods

**Mice and parasites**. This study was carried out in accordance with the recommendations in the Guide for the Care and Use of Laboratory Animals of the National Institutes of Health. The protocol was approved by the Institutional Animal Care and Use Committee of New York University School of Medicine, which is fully accredited by the Association for Assessment and Accreditation of Laboratory Animal Care International.

Female Swiss Webster mice 6–8 weeks old were purchased from The Jackson Laboratory (Bar Harbor, ME). For transgenic mice models, age matched uMT, (stock# 002288) *Ifngr1*[−/−] (stock# 003288) and *Tbx21*[−/−] (stock# 004648) mice on the C57BL/6 (stock# 000664) background were purchased from The Jackson Laboratory (Bar Harbor, ME, USA). *Tlr9*[−/−] mice were a kind gift from Dr. G. Miller, which he acquired from S. Akira (Osaka University, Osaka, Japan) and have been described previously[44]. All mice were kept in a standard barrier animal facility. All mice were euthanized following standard $CO_2$ chamber procedures.

To start blood-stage infections (BSL1), at least 5 mice per group for every experiment, were injected i.p. with $1 \times 10^6$ infected RBCs per mouse of the nonlethal strain *P. yoelii* 17XNL resuspended in PBS, final volume 250 µl. To evaluate parasitemia, thin blood smears were made by bleeding mice from a nick in the tail. Smears were stained with KaryoMAX Giemsa (Life Technologies, Norwalk, CT), and a minimum of 500 RBCs per smear were counted. To evaluate anaemia, RBC numbers were counted in Neubauer Chamber in an inverted light-microscope. *P. yoelii* 17XNL-infected RBCs were harvested by cardiac puncture of infected, anesthetized mice before the peak of parasitemia. RBCs were washed twice with PBS and separated from white blood cells by centrifugation at $2000 \times g$ for 3 min. RBCs were then spun on an Accudenz (Accurate Chemical and Scientific, Westbury, NY) gradient to isolate schizonts and late trophozoite stage-infected RBCs, washed, and resuspended in PBS. *P. yoelii* and *P. falciparum*-infected RBCs were isolated through LD magnetic columns (Miltenyi) and infected or uninfected erythrocytes were lysed by freeze–thaw 10 times with liquid nitrogen and kept sterile.

***P. falciparum* culture and isolation**. Erythrocyte asexual stage cultures of the *P. falciparum* strain 3D7 were maintained at 5% hematocrit in RPMI 1640, 25 mM HEPES supplemented with 10 µg/ml gentamicin, 250 µM hypoxanthine, 25 mM sodium bicarbonate, and 0.5% Albumax II (pH 6.75) under atmospheric conditions of 5% oxygen, 5% carbon dioxide, and 95% nitrogen. Magnetic separation of late stages with MACS cell separation columns (Miltenyi Biotec) was used for culture

synchronization and to isolate late-stage infected RBC for use in experiments. For experiments with lysates, late-stage infected RBC were lysed by 10 freeze/thaw cycles. *P. falciparum* culture supernatants were tested for mycoplasma contamination using the MycoAlert Mycoplasma Detection Kit (Lonza LT07-118) and found to be negative.

**Human PBMC enrichment**. Peripheral venous blood from healthy malaria-naive donors was obtained on the day of the experiment at the New York University Clinical and Translational Science Institute with sodium citrate as anticoagulant. Institutional Review Board (IRB) approval was obtained at New York University School of Medicine. PBMCs were enriched using Ficoll-Paque PLUS (GE life sciences). All recruited volunteers provided written informed consent prior to blood donation.

**Flow cytometry**. All flow cytometry was performed on a FACSCalibur (Becton Dickinson, Franklin Lakes, NJ) and analyzed with FlowJo (Tree Star, Ashland, OR). Most Abs for FACS were purchased from BioLegend (San Diego, CA) except the following: anti-MHC class II from BD Pharmigen (San Jose, CA) and anti-CD27 and anti-CD32b from eBioscience (San Diego, CA). To characterize our mouse CD11c$^+$T-bet$^+$B cells all antibodies were used at 1 μg/ml. The following antibodies were used in FITC: anti-CD19 (6D5), anti-CD23 (B3B4), anti-CD27 (LG.3A10), anti-MHC class II (M5/114.15.2), anti-CD1d (1B1), and anti-CD5 (53-7.3); in PE: anti-CD22 (OX-97), anti-CD138 (281-2), anti-CD80 (16-10A1), anti-CR1/CR2 (CD35/CD21) (7E9), anti-T-bet (4B10), anti-CD11b (M1/70), and anti-CD24 (M1/69); in PRCP: anti-B220 (RA3-6B2); and in APC: anti-CD11c (N418). All antibodies were matched with their correct fluorescent-conjugated isotype control. For human assays, PBMCs were stained with FITC anti-CD20 (2H7), PE anti-T-bet (4B10), PRCP anti-CD11c (3.9), and APC anti-CD19 (HIB19). Intracellular T-bet staining was performed using the True-Nuclear™ Transcription Factor Buffer Set (Biolegend) and following manufacturer's instructions.

**ELISA**. Costar 3590 ELISA plates were coated with PS at 20 μg/ml or uninfected RBC lysate ($10^9$ RBC/ml in PBS) diluted 1:500 in 200 proof Molecular Biology ethanol and allowed to evaporate at RT after >16 h of incubation at 4 °C. Plates were washed 5 times with PBS 0.05% tween 20 and then blocked for 1 h with PBS 3% BSA. Plasma from mice was diluted at 1:100 in blocking buffer and incubated for 2 h at 37 °C. Plates were washed again 5 times and incubated with anti-mouse IgG-HRP (GE Healthcare) for 1 h at 37 °C. Plates were washed 5 more times and TMB substrate (BD Biosciences) was added until desired color was obtained. Reaction was stopped with Stop buffer (Biolegend) and absorbance was read at 450 nm. The mean OD at 450 nm from triplicate wells was compared with the same dilution of a reference positive serum to calculate relative units (RU). For human PBMCs ELISAs, a similar process was done but using human RBC lysates or PS for coating, undiluted PBMCs culture supernatants and anti-human IgM-HRP (Millipore) for detection.

**ELISPOT assay**. Mouse ELISPOTs were done as previously reported[14]. Mouse splenocyte suspensions from uninfected control and *P. yoelii*-infected mice (day 6 post infection) were obtained. White blood cells were additionally purified in 45% Percoll at 25 °C for 20 min at 1500 × *g* to remove remaining RBCs. Specific B-cell populations were enriched through magnetic bead sorting (Miltenyi) by positive selection with anti-CD11c (atypical) or anti-CD138 (plasma cell) coated magnetic beads. Purification yield was assessed by flow cytometry prior to addition to plate (Supplementary Fig. 1). $5 \times 10^4$ cells were added per well and incubated in RPMI 1640 supplemented with 10% FBS in 96-well Costar 3590 ELISA plates (Corning Life Sciences, Tewksbury, MA) precoated with either capture anti-IgG (15 mg/ml), uninfected RBC lysate, PS (100 mg/ml in ethanol) (Sigma, St. Louis, MO), or PBS 10% BSA as control for 20 h at 37 °C with 5% CO2. Following extensive washings, anti-mouse IgG biotinylated detection antibody (Sigma, St. Louis, MO) was added at 1 μg/ml diluted in PBS 0.5% FBS for 2 h at RT. Streptavidin-horseradish peroxidase (Mabtech AB, Nacka Strand, Sweden) was added diluted in PBS 0.5% FBS for 1 h at RT. Plates were developed with TMB substrate (Mabtech AB, Nacka Strand, Sweden) for 15–20 min and then washed extensively with water. Spots were quantified by microscopy.

**In vitro activation assays**. Naive B cells from mice were purified as a CD43 negative fraction using anti-CD43 beads (Miltenyi) and confirmed by flow cytometry to express CD19 (>96%). Cells were cultured at $5 \times 10^6$ cells per ml for 2 or 3 days as indicated. TLR7 agonist R848 (Mabtech) was used at 1 μg/ml and TLR9 agonist CpG (InvivoGen) at 1 μg/ml; anti-BCR (Fab')$_2$ anti-IgM (The Jackson Laboratory) was used at 5 μg/ml, and IFN-γ (Peprotech) at 100 U/ml. Uninfected and infected lysates were added at a concentration of 1:10 (B-cell: RBC), as measured before lysis. TLR7 antagonist OD 20958 (Miltenyi) and TLR9 antagonist ODN A151 (InvivoGen) were used at 4 and 8 μg/ml, respectively. For DNase/RNase treatments: Lysate fractions were treated with 4 units of DNase (TURBO DNase, Life Technologies), 4 units of RNase A (Invitrogen) or mock treated with a similar volume of DNase buffer or PBS. All treatments were incubated at 37 °C for 16 h, and kept sterile.

Human PBMCs were seeded in flat 96-wells at a density of $2.5 \times 10^4$ per well. *P. falciparum*-infected RBC lysates were prepared as mentioned before and added at a ratio of 1:10 (PBMCs:RBC) and cultured in serum-free hematopoietic cell X-VIVO 15 media (Lonza) for 6 days (flow cytometry/ IFN-γ quantification) or 10 days (supernatant for ELISA). TLR antagonists were added at the same concentrations as for mouse B-cell assays. IFN-γ levels were quantified from culture supernatants with BD Cytometric Bead Arrays (BD Biosciences) Human Th1/Th2 Cytokine Kit. The arrays were acquired on a FACScalibur (BD Biosciences) and analyzed with FCAP Array Software (BD Biosciences).

**Mixed bone marrow chimeras**. Chimeric mice harboring either WT, uMT, $Tbx21^{-/-}$, or $Ifngr1^{-/-}$ B cells were generated as previously described[31]. For example, to generate mice with $Tbx21^{-/-}$ B cells, bone marrow ($1 \times 10^6$ cells) from CD45.2 $Tbx21^{-/-}$ donor mice was mixed with bone marrow ($9 \times 10^6$ cells) from CD45.2 B-cell-deficient donor mice (uMT) and $1 \times 10^7$ total bone marrow cells (1:9 ratio) were injected i.v. into irradiated recipients. Recipients were WT mice (CD45.1) that were irradiated with 6.5 and 5.5 Gy with a$^{137}$Cs Nordian GammaCell 40 Exactor, separated by 12 h, prior to bone marrow transfer. To generate chimeric mice with $Ifngr1^{-/-}$ B cells, or control chimeric mice with WT B cells, identical procedures were followed using bone marrow from either $Ifngr1^{-/-}$ or WT donors mixed with uMT bone marrow. All chimeric mice were maintained on oral sulfamethoxazole for 2 weeks after irradiation (6.5 and 5.5 Gy separated by 12 h). Chimerism was assessed at 6 weeks and mice were infected with *P. yoelii* at 8 weeks. For B-cell-specific deletion of specified genes, chimerism for all mice was >80%.

**Statistical analysis**. All analyses were performed using GraphPad Prism version 5.0 (GraphPad Software, La Jolla, CA). Statistical significance of differences between two groups was assessed by unpaired Student's *t* tests or ANOVA as indicated. A *p* value of <0.05 was considered significant. Error bars represent the standard deviations (SD) of the total of mice used per experiment.

**Data availability**. All relevant data are available from the authors upon request.

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

## Acknowledgements

This work was supported in part by the National Institutes of Health (NIH) institutional training grants 5T32AI100853-03 and 5T32AI007180 to J.R.C. and R01AI125446 and R01AI127481 to N.S.B. Healthy donors blood draw was performed at NYU CTSI, which is supported by grants 1UL1TR001445, 1KL2 TR001446, and 1TL1 TR001447 from the National Center for the Advancement of Translational Science (NCATS), NIH. We acknowledge Dr. Anton Götz, Maureen Ty and Marisol Zuniga from the Rodriguez Lab at NYU for their help with human PBMC isolation and parasite culture. We also thank Dr. G. Miller from NYU for providing us with the Tlr9$^{-/-}$ mice. The content is solely the responsibility of the authors and does not necessarily represent the official views of the National Institutes of Health.

## Author contributions

J.R.C., R.V., and J.J.G. designed, performed, and analyzed data from the manuscript. C.F.-A., M.A.P.-R., N.S.B., and S.G. provided scientific support, assistance in discussing and interpreting experimental results and related assays. J.R.C., N.S.B., and A.R. conceived the experiments, analyzed the data, and wrote the manuscript. All the authors have read and commented on the final manuscript and have agreed to its submission.

## Additional information

**Competing interests:** The authors declare no competing financial interests.

