## [Peer Review File · Nature Communications]

Reviewers' comments:

Reviewer #1 (autoreactive B, IFN γ)(Remarks to the Author):

In this manuscript Rivera-Correa et al. show that Plasmodium induces the appearance of T-bet⁺ B cells (ABCs) via TLR9 and IFN γ R engagement, leading to the development of anti-RBC autoantibodies and anemia.

Overall this research topic is novel and important since several groups have already demonstrated the involvement of T-bet⁺ B cells in viral clearance and in the development and/or progression of autoimmunity. However, the appearance of this type of B cells has not previously been reported for infections such as malaria, so this MS will be important to the field. The authors demonstrate that CD11c⁺/T-bet⁺ B cells accumulate upon Plasmodium infection (day 9). The phenotype of these cells recapitulates that of the previously described phenotype of ABCs. In addition, these CD11c⁺ B cells are enriched in autoreactive anti-RBC cells, suggesting their role in infection-induced anemia. The authors demonstrate that the induction of these cells is driven by malaria infection itself, since the authors show that infected lysates induce T-bet expression in uninfected B cells in the presence of IFN γ and anti-IgM in TLR9 dependent manner.

However, there are number of issues which dampen enthusiasm for this MS. Problems involve in particular the experiments using IFN γ R^{-/-}, T-bet^{-/-} and TLR9^{-/-} mice:

Major comments:

1. The authors should state how many times the experiments have been repeated.
2. In Figures 5 and 6 the authors claim that T-bet, IFN γ R and TLR9 deficiency leads to resistance to malaria-induced anemia, suggesting that the anemia is mediated by the CD11c⁺ Tbet⁺ B cells (page 9). This statement is not supported by the data presented in the manuscript since the experiments involved mice that were complete knock-outs for the relevant genes. Thus the effects could have been mediated by other cell types that also normally express IFN γ R, T-bet or TLR9. To confirm that the effect is B cell intrinsic, B cell specific knock out animals should be used (or mixed BM chimeras have to be generated such that only the B cells lack the relevant genes –for example chimeras reconstituted with a mixture of uMT and IFN γ R^{-/-} bone marrow cells.
3. On Pg 6 the authors write, “ It is well characterized that intracellular expression of T-bet, a reliable marker for autoreactive B cells (27), is highly dependent on interferon gamma signaling (28)”. This is a misleading statement since T-bet⁺ B cells can be found during various viral infections (Rubtsova, PNAS, 2013) and T-bet⁺ memory B cells have been previously described (Wang, Nat. Immunology, 2012), therefore, T-bet expression is not truly “a reliable marker for autoreactive B cells” or at least, not only for autoreactive B cells (reference #27 does not support this statement either).

Minor comments:

1. In Fig 3 authors present the data as MFI of T-bet expression in B cells, however, later in Figs 4 and 7, they switch to % of T-bet⁺ B cells. In this case (when the entire population shifts in T-bet expression) the MFI is the best way to present the data. I'd recommend that the authors change the way they present the data in Fig 4 and 7 to make it consistent with Fig 3.
2. Fig4 ODN 20958 and ODN A151 are not the commonly used names for TLR7 and TLR9 antagonists, I suggest the authors include the words “TLR7/9 antagonist” to make it easier to understand.
3. Fig 5. As mentioned above, these data are weak, because the B cell specific knock out has to be used to reach the conclusions the authors would like to make. In addition, the authors should report the data for day 8-9 after infection, because, as shown in Fig 1 these are the days on which the numbers of CD11c⁺/T-bet⁺ B cells peak. Thus the absence of CD11c⁺/T-bet⁺ B cells should have the most dramatic effect during days 8-9.

4. Fig 6b-c - How do authors explain the fact that anti-PS antibodies are comparable between WT and Tlr9-/- at days 12-15, but the PBC counts are increased in TLR9-/- mice ?
5. Fig 7b. Again, in this figure the data are incorrectly represented. The data should be presented as the MFI of T-bet in B cells, or as the % of B cells that are T-bet+, not the % of all cells that are T-bet+. This is because the % of B cells changes dramatically (from 10 to 30%) between the various mice.
6. Fig 7d,e – The concentrations of the antibodies should be presented as titers not as OD, measured by titrations.

Reviewer #2 (auto-Ab, autoimmunity)(Remarks to the Author):

induced autoimmunity. This approach was conducted to show that T-bet B cells is mediated directly through intrinsic TLR9 recognizing Plasmodium DNA along with IFN γ that synergizes to promote expansion of mouse and human Tbet B cells and generate autoantibodies. These data are in contrast to Butler et al ,J Immunol 2017, which show that B cell–intrinsic T-bet expression impairs GC B cell differentiation and limits GC B cell derived Ab responses during experimental malaria. Butler claims that B cell-intrinsic IL-10 enhances whereas B cell-intrinsic IFN- γ and T-bet suppress GC B cell responses and anti-Plasmodium humoral immunity. IL-10 also indirectly supports humoral immunity by suppressing excessive IFN-g, which induces T-bet expression in B cells. It is hard to get into conclusion in such an isolated system of TLRs. The study can be published with major corrections.

Major points:

1. All over the manuscripts there are a lot of general statements which are not based on the results.
2. The title should be more soft according to the results. Plasmodium do not cause autoimmunity or autoimmune disease, but autoimmune like condition.
3. TLR9 is not the only mechanism to explain infection-induced autoimmunity.
4. The authors did not show in the past and nor in the current paper the response to plasmodium itself which encompass anti-PS , just to RBCs which also express PS on non infected RBCs. The antiphosphatidylserine (PS) generation is part of an innate response to plasmodium. Anti-plasmodium, anti-dsDNA, anti-phosphatidylcholine which characterize malaria are neglected.
5. PS is a part of phospholipid biosynthesis pathways in Plasmodium falciparum. Its immunogenicity can be related to PS-carrier structure due to response to Plasmodium. Yes, it bind also the PS on RBCs and apoptotic cells and accelerate their clearance by macrophages like in antiphospholipid syndrome and others autoimmune diseases.
6. T bet B cell population is not the only B cells producing autoantibodies.
7. In the authors hands, malaria induce the Tbet B cell via a novel pathway involving IFN γ R, BCR and TLR9. Its in debate in the literature, but the current selected data confirm this pathway.

Reviewer #3 (malaria, TLR, innate)(Remarks to the Author):

Recent studies by Dr. Rodriguez group have elaborated on the role of anti-phosphatidylserine (PS) and anti-erythrocyte (RBC) antibodies as important elements on malaria-induced anemia. Their study now indicates that B lymphocytes which produce anti-PS/antiRBC antibodies are a unique subset of CD11c+CD11b+T-bet+B cells. In addition, they report that induction of these antibodies is dependent on IFN γ , TLR9 and T-Bet. This subset of B-lymphocytes has been studied in detail by Dr. Marrack's group and its importance both on production neutralizing antibodies in viral infection and autoantibodies has been demonstrated. Hence, the main novelty here is the demonstration of the role

of these cells in anemia induced by Plasmodium infection in mice. Furthermore, they report that B cells from peripheral blood mononuclear cells (PBMCs) stimulated with infected RBCs transform into T-bet⁺ B lymphocytes. It is unfortunate that the levels of circulating anti-PS antibodies persist high in either IFNG^{-/-} or T-bet^{-/-} mice infected with Plasmodium. This finding begs the question of what is the alternative source of these anti-erythrocyte antibodies.

Experiments requested to improve the relevance of the study:

- 1) A key experiment is the use of conditional t-Bet KO mice in B lymphocytes. This mouse lineage has just been published (P. Maracka et al. JCI 2017) and should be available for experiments. This experiment would more definitely show that T-bet⁺ B lymphocytes are the critical source of anti-PS antibody production in vivo and malaria anemia and highly improve the manuscript.
- 2) The human experiments are a plus in this study. The demonstration of an enhanced frequency of T-bet⁺ B cells in PBMCs from malaria patients would improve the relevance of the findings in human disease. Also, the demonstration that purified T-bet⁺ vs conventional B cells from malaria patients make anti-PS antibodies is a key demonstration for the relevance of these cells in malaria patients.
- 3) Another interesting point is the work done by P. Marracka lab with T-bet⁺ cells was on an experimental lupus model, where anti-DNA/RNA are the main effector antibodies. A recent study performed by B. Diamond and colleagues (e.g., mBio 2015;6:e01605-15) demonstrates that high levels of anti-DNA antibodies are found in the circulation of malaria patients. Demonstration that T-bet cells from malaria patients are also the source of anti-DNA abs would allow more broad conclusions. I understand that this is a bit beyond Dr. Rodriguez interest. In any case, it should be included in the discussion.
- 4) The effects of TLR9 and TLR7 antagonists on B cell differentiation are minor. However, there is a major redundancy between these two receptors and different studies show that both Plasmodium DNA and RNA activated innate immune receptors. I suggest that new in vitro experiments, similar to the one presented in Figure 4, are performed using both antagonist (for TLR7 and TLR9) in the same condition.
- 5) Use of MyD88^{-/-} mice or 3D mice (deficient in nucleic acid sensing TLRs) may be very helpful, both in vivo and in vitro, to evaluate the question of redundancy of TLR7/TLR9 and may be other TLRs.

Suggestions to improve manuscript presentation:

- 6) There is a vast literature on TLR9 activation by parasite DNA as well as IFN γ and their role on innate and acquired immune responses during malaria. This should be acknowledged and brought to the discussion.
- 7) A panel showing expression of CD138 by this unique subset versus conventional B lymphocytes should be presented in Figure 1. Better, Figure 1 should illustrate not only differences between infected vs uninfected mice, but also needs a panel that shows the difference on expression of critical markers when comparing conventional and the CD11c B cell subset. In panel 1E, the levels of anti-PS vary a lot. I am wondering if this variation associates with anemia. It does not seem to be so, as RBC count in infected mice does not vary too much from mice to mice.

Point-by-point response to reviewers

Reviewer 1:

In this manuscript Rivera-Correa et al. show that Plasmodium induces the appearance of T-bet⁺ B cells (ABCs) via TLR9 and IFN γ R engagement, leading to the development of anti-RBC autoantibodies and anemia.

Overall this research topic is novel and important since several groups have already demonstrated the involvement of T-bet⁺ B cells in viral clearance and in the development and/or progression of autoimmunity. However, the appearance of this type of B cells has not previously been reported for infections such as malaria, so this MS will be important to the field. The authors demonstrate that CD11c⁺/T-bet⁺ B cells accumulate upon Plasmodium infection (day 9). The phenotype of these cells recapitulates that of the previously described phenotype of ABCs. In addition, these CD11c⁺ B cells are enriched in autoreactive anti-RBC cells, suggesting their role in infection-induced anemia. The authors demonstrate that the induction of these cells is driven by malaria infection itself, since the authors show that infected lysates induce T-bet expression in uninfected B cells in the presence of IFN γ and anti-IgM in TLR9 dependent manner.

However, there are number of issues which dampen enthusiasm for this MS. Problems involve in particular the experiments using IFN γ R^{-/-}, T-bet^{-/-} and TLR9^{-/-} mice:

Major comments:

1. The authors should state how many times the experiments have been repeated.

The times each experiment was repeated is now included in every figure legend. Every experiment has been repeated at least 2 times, except for the quantification of parasitemia, anemia and anti-PS antibodies in infected mice (Fig. 1d-f) where 45 total mice were sacrificed in groups of 3 for every time point of the experiment. The human data were obtained with cells from 5 different donors in individual experiments performed in different days.

2. In Figures 5 and 6 the authors claim that T-bet, IFN γ R and TLR9 deficiency leads to resistance to malaria-induced anemia, suggesting that the anemia is mediated by the CD11c⁺ Tbet⁺ B cells (page 9). This statement is not supported by the data presented in the manuscript since the experiments involved mice that were complete knock-outs for the relevant genes. Thus the effects could have been mediated by other cell types that also normally express IFN γ R, T-bet or TLR9. To confirm that the effect is B cell intrinsic, B cell specific knock out animals should be used (or mixed BM chimeras have to be generated such that only the B cells lack the relevant genes –for example chimeras reconstituted with a mixture of uMT and IFN γ R^{-/-} bone marrow cells.

The reviewer is correct that our original approach did not allow us to determine if the effects of IFN γ R, T-bet or TLR9 were B cell intrinsic. Thus, following the reviewer suggestion, we have first analyzed the levels of anti-PS antibodies in mixed bone marrow chimeric mice in which either IFN γ R^{-/-} or T-bet deficiency is restricted to the B cell compartment. These chimeras had been

generated in Dr. Noah Butler's group and published before (Guthmiller et al., *J. Immunol.* 2017 Jan 15;198(2):617-622).

We found that the levels of anti-PS antibodies generated during *P. yoelii* infection were lower in these chimeric mice compared to control chimeric mice harboring WT B cells. Importantly, parasite (MSP1)-specific antibody responses are actually elevated in chimeric mice with B cells lacking either $IFN\gamma R$ or $T\text{-bet}$, suggesting that the lack of expression of $IFN\gamma R$ or $T\text{-bet}$ in B cells selectively affects the production of anti-PS antibodies (but not the quantity or quality of germinal center-derived parasite-specific antibodies). These results have been included in new Fig. 6.

In order to address the role of TLR-9 in the generation of an anti-PS response, we generated 10 mixed bone marrow chimeric mice with B cells lacking TLR-9. Unexpectedly, chimeric mice with $TLR9^{-/-}$ B cells harbored 10-fold fewer peripheral mature B cells, compared to control chimeras with WT B cells. Thus, at least in this context, TLR9 deficiency has a marked impact on the development and differentiation of B cells. Despite these differences in total numbers of B cells among WT and TLR9 mixed bone marrow chimeras, we infected control and chimeric mice with *P. yoelii* and measured PS- and parasite-specific secreted antibody responses using ELISA and ELISpot. The new data show that although the levels of anti-PS antibodies were not significantly different in the TLR-9 $^{-/-}$ B cell chimeric mice when compared with controls, we observed a significant reduction in the levels of anti-PS secreting B cells by ELISpot (see graphs below).

B-cell specific deletion of TLR9 leads to a reduced anti-PS B-cell response during *P. yoelii* infection.

(a) Quantification of the ratio of anti-PS IgG versus anti-*P.yoelii* MSP1 antibodies (RU, Relative Units) from B-cell-specific chimeras reconstituted with either WT (circle) or $TLR9^{-/-}$ B cells (squares). (b) Number of antigen-specific spots for PS over parasite *P.yoelii* MSP1 of B-cell specific $TLR9^{-/-}$ chimeric mice at D.18 p.i. Significance determined by unpaired Student's *t* test * $p < 0.05$. Graphs represent the individual values of $n=11$ mice (a) or $n=4$ mice (b) *** $p < 0.001$

Results are expressed as a ratio of anti-PS versus anti-MSP1 IgG. Similar results were found when data are expressed simply as anti-PS IgG antibodies. Even though these new ELISpot data

support our model and hypotheses, given the caveats associated with the unequal B cell reconstitution in these chimeric mice, we have elected not to include these data in the revised manuscript.

3. On Pg 6 the authors write, “ It is well characterized that intracellular expression of T-bet, a reliable marker for autoreactive B cells (27), is highly dependent on interferon gamma signaling (28)”. This is a misleading statement since T-bet⁺ B cells can be found during various viral infections (Rubtsova, PNAS, 2013) and T-bet⁺ memory B cells have been previously described (Wang, Nat. Immunology, 2012), therefore, T-bet expression is not truly “a reliable marker for autoreactive B cells” or at least, not only for autoreactive B cells (reference #27 does not support this statement either).

This has been corrected in the text.

Minor comments:

1. In Fig 3 authors present the data as MFI of T-bet expression in B cells, however, later in Figs 4 and 7, they switch to % of T-bet⁺ B cells. In this case (when the entire population shifts in T-bet expression) the MFI is the best way to present the data. I'd recommend that the authors change the way they present the data in Fig 4 and 7 to make it consistent with Fig 3.

The data presented in Figs. 3 and 4 shows the T-bet levels of purified B cells. We have modified the analysis for consistency and chose to express the results as % because the entire population does not shift in a uniform way. Figure 7 (now figure 8) has also been analyzed as % of total B cells.

2. Fig4 ODN 20958 and ODN A151 are not the commonly used names for TLR7 and TLR9 antagonists, I suggest the authors include the words “TLR7/9 antagonist” to make it easier to understand.

The names of the TLR inhibitors have been modified throughout the manuscript and in the figures following this suggestion.

3. Fig 5. As mentioned above, these data are weak, because the B cell specific knock out has to be used to reach the conclusions the authors would like to make. In addition, the authors should report the data for day 8-9 after infection, because, as shown in Fig 1 these are the days on which the numbers of CD11c⁺/T-bet⁺ B cells peak. Thus the absence of CD11c⁺/T-bet⁺ B cells should have the most dramatic effect during days 8-9.

These experiments were repeated twice, once the levels of CD11c⁺/T-bet⁺ B cells were measured in day 8 and in the second experiment in day 10. Both resulted in a similar difference in levels of CD11c⁺/T-bet⁺ B cells between wt and ko mice.

4. Fig 6b-c - How do authors explain the fact that anti-PS antibodies are comparable between WT and Tlr9^{-/-} at days 12-15, but the PBC counts are increased in TLR9^{-/-} mice ?

It is likely that the effect of anti-PS antibodies in clearing RBC is observed for a few days (2 to 5 days), which is the time an increase in erythropoiesis will take to produce new RBC to replace the

lost ones. In fact, the difference is only observed for 3 days (from day 9 to day 12) since it is no longer significant by day 15.

5. Fig 7b. Again, in this figure the data are incorrectly represented. The data should be presented as the MFI of T-bet in B cells, or as the % of B cells that are T-bet+, not the % of all cells that are T-bet+. This is because the % of B cells changes dramatically (from 10 to 30%) between the various mice.

We specially thank the reviewer for this suggestion since the re-analysis of the TLR9 antagonist data to make them relative to number of total B cells and not total number of cells showed that this antagonist interferes with the entire B cell population, not only with Tbet positive B cells. This is similar to our findings with the reduced number of total B cells in TLR9 B cell chimeric mice (point 2 above), where it appears that TLR9 is necessary for the entire B cell population. These results suggest an unexpected role of TLR9 in B cell development and/or survival that should be addressed in a separate study.

Former Fig. 7 (now Fig. 8) has been modified to remove the antagonist data on human PBMC. We still consider valid the effect of the TLR9 antagonist in mouse purified B cells, since in this case the quantification is referred to total B cells (Figure 4).

6. Fig 7d,e – The concentrations of the antibodies should be presented as titers not as OD, measured by titrations.

The production of antibody levels in culture supernatants in vitro leads to low concentrations of antibodies in the supernatants. In our preliminary experiments, we observed that dilutions of the supernatants as low as 1:4 already were undetectable by ELISA.

Reviewer 2:

induced autoimmunity. This approach was conducted to show that T-bet B cells is mediated directly through intrinsic TLR9 recognizing Plasmodium DNA along with IFN γ that synergizes to promote expansion of mouse and human Tbet B cells and generate autoantibodies. These data are in contrast to Butler et al ,J Immunol 2017, which show that B cell–intrinsic T-bet expression impairs GC B cell differentiation and limits GC B cell derived Ab responses during experimental malaria. Butler claims that B cell-intrinsic IL-10 enhances whereas B cell-intrinsic IFN- γ and Tbet suppress GC B cell responses and anti-Plasmodium humoral immunity. IL-10 also indirectly supports humoral

immunity by suppressing excessive IFN- γ , which induces T-bet expression in B cells. It is hard to get into conclusion in such an isolated system of TLRs. The study can be published with major corrections.

The reviewer raises relevant and important questions about the role of IFN- γ and T-bet in shaping humoral immunity during Plasmodium infection.

We respectfully contend that our new data are perfectly consistent with the data sets recently published in the Journal of Immunology (Guthmiller et al. JI. 2017). Indeed, in every genetic manipulation that results in enhanced germinal center reactions and elevated (high quality) anti-parasite (MSP1)-specific antibody responses, we see a contemporaneous and marked reduction in “non-parasite-specific antibody responses (including PS-specific responses) and pathologic B cells reactions. For example, in the JI paper, we found that specific deletion of T-bet and IFN γ R signaling in B cells enhances high quality, parasite-specific antibody responses. Importantly, in the new data presented here in our revised manuscript, these same genetic manipulations (deletion of T-bet and IFN γ R signaling in B cells) result in reduced anti-PS responses.

Major points:

1. All over the manuscripts there are a lot of general statements which are not based on the results.

The manuscript has been reviewed carefully to remove any conclusions that were not solidly supported by the results.

2. The title should be more soft according to the results. Plasmodium do not cause autoimmunity or autoimmune disease, but autoimmune like condition.

The title and abstract have been modified to reflect that this is an autoimmune-like condition.

3. TLR9 is not the only mechanism to explain infection-induced autoimmunity.

*We agree with the reviewer and have revised the manuscript to clarify that this is a mechanism that can **contribute** to autoreactivity.*

4. The authors did not show in the past and nor in the current paper the response to plasmodium itself which encompass anti-PS , just to RBCs which also express PS on non infected RBCs. The

antiphosphatidylserine (PS) generation is part of an innate response to plasmodium. Anti-plasmodium, anti-dsDNA, anti-phosphatidylcholine which characterize malaria are neglected.

5. PS is a part of phospholipid biosynthesis pathways in *Plasmodium falciparum*. Its immunogenicity can be related to PS-carrier structure due to response to Plasmodium. Yes, it bind also the PS on RBCs and apoptotic cells and accelerate their clearance by macrophages like in antiphospholipid syndrome and others autoimmune diseases.

Response to points 4 and 5: We agree with the reviewer that it remains unknown whether the PS derived from Plasmodium merozoites or the PS derived from host RBC is the antigen triggering the anti-PS response. However, in any case, the resulting antibodies are autoreactive. In this work we are focusing on identifying the anti-PS producing B cells and the mechanism leading to their activation during malaria.

IgM naturally occurring antibodies recognizing mainly phosphatidylcholine and malondialdehyde are found at low levels in healthy subjects. However, our work is focused on malaria-induced IgG anti-PS antibodies, which are not considered innate-like or natural antibodies (Elkon and Silverman, in “Naturally Occurring Antibodies” chapter 2, 2012).

Regarding other specificities of antibodies: anti-MSP1 antibodies were measured and compared with anti-self antibodies in our previous publication (Fernandez-Arias et al., Cell Host Microbe, 2016), anti-DNA antibodies are shown in this current manuscript and new data including anti-phosphatidylcholine and other phospholipids in Supplemental figure 2. An ELIspot showing B cells specific for DNA and for different phospholipids is also shown here. Whether the antibody reactivity to different lipids is due to cross-reactive antibodies or to the generation of antibodies to each individual lipid is not clear. It is likely that both phenomena are responsible for the observed results.

The focus of this manuscript is the anti-PS antibodies because of their pathogenic role in malaria-induced anemia.

6. T bet B cell population is not the only B cells producing autoantibodies.

We agree with the reviewer statement and this has been corrected in the manuscript.

7. In the authors hands, malaria induce the Tbet B cell via a novel pathway involving IFN γ R, BCR and TLR9. Its in debate in the literature, but the current selected data confirm this pathway.

We have modified the text according to reviewer suggestion.

Reviewer 3:

Recent studies by Dr. Rodriguez group have elaborated on the role of anti-phosphatidylserine (PS) and anti-erythrocyte (RBC) antibodies as important elements on malaria-induced anemia. Their study now indicates that B lymphocytes which produce anti-PS/antiRBC antibodies are a unique subset of CD11c+CD11b+T-bet+B cells. In addition, they report that induction of these antibodies is dependent on IFN γ , TLR9 and T-Bet. This subset of B-lymphocytes has been studied in detail by Dr. Marrack's group and its importance both on production neutralizing antibodies in viral infection and autoantibodies has been demonstrated. Hence, the main novelty here is the demonstration of the role of these cells in anemia induced by Plasmodium infection in mice. Furthermore, they report that B cells from peripheral blood mononuclear cells (PBMCs) stimulated with infected RBCs transform in T-bet+ B lymphocytes. It is unfortunate that the levels of circulating anti-PS antibodies persist high in either IFNG $^{-/-}$ or T-bet $^{-/-}$ mice infected with Plasmodium. This finding begs the question of what is the alternative source of these anti-erythrocyte antibodies.

Experiments requested to improve the relevance of the study:

1) A key experiment is the use of conditional t-Bet KO mice in B lymphocytes. This mouse lineage has just been published (P. Maracka et al. JCI 2017) and should be available for experiments. This experiment would more definitely show that T-bet+ B lymphocytes are the critical source of anti-PS antibody production in vivo and malaria anemia and highly improve the manuscript.

Following the reviewer suggestion, we have first analyzed the levels of anti-PS antibodies in mixed bone marrow chimeric mice in which either IFN γ R $^{-/-}$ or T-bet deficiency is restricted to the B cell compartment. These chimeras had been generated in Dr. Noah Butler's group and published before (Guthmiller et al., J. Immunol. 2017 Jan 15;198(2):617-622).

We found that the levels of anti-PS antibodies generated during P. yoelii infection were lower in these chimeric mice compared to control chimeric mice harboring WT B cells. Importantly, parasite (MSP1)-specific antibody responses are actually elevated in chimeric mice with B cells lacking either IFN γ R or T-bet, suggesting that the lack of expression of IFN γ R or T-bet in B cells selectively affects the production of anti-PS antibodies (but not the quantity or quality of germinal center-derived parasite-specific antibodies). These results have been included in new Fig. 6.

In order to address the role of TLR-9 in the generation of an anti-PS response, we generated 10 mixed bone marrow chimeric mice with B cells lacking TLR-9. Unexpectedly, chimeric mice with

*TLR9^{-/-} B cells harbored 10-fold fewer peripheral mature B cells, compared to control chimeras with WT B cells. Thus, at least in this context, TLR9 deficiency has a marked impact on the development and differentiation of B cells. Despite these differences in total numbers of B cells among WT and TLR9 mixed bone marrow chimeras, we infected control and chimeric mice with *P. yoelii* and measured PS- and parasite-specific secreted antibody responses using ELISA and ELISpot. The new data show that although the levels of anti-PS antibodies were not significantly different in the TLR9^{-/-} B cell chimeric mice when compared with controls, we observed a significant reduction in the levels of anti-PS secreting B cells by ELISpot (see graphs below).*

B-cell specific deletion of TLR9 leads to a reduced anti-PS B-cell response during *P. yoelii* infection.

(a) Quantification of the ratio of anti-PS IgG versus anti- *P.yoelii* MSP1 antibodies (RU, Relative Units) from B-cell-specific chimeras reconstituted with either WT (circle) or TLR9^{-/-} B cells (squares). (b) Number of antigen-specific spots for PS over parasite *P.yoelii* MSP1 of B-cell specific TLR9^{-/-} chimeric mice at D.18 p.i. Significance determined by unpaired Student's *t* test **p* < 0.05. Graphs represent the individual values of n=11 mice (a) or n=4 mice (b) ****p* < 0.001

Results are expressed as a ratio of anti-PS versus anti-MSP1 IgG. Similar results were found when data are expressed simply as anti-PS IgG antibodies. Even though these new ELISpot data support our model and hypotheses, given the caveats associated with the unequal B cell reconstitution in these chimeric mice, we have elected not to include these data in the revised manuscript.

2) The human experiments are a plus in this study. The demonstration of an enhanced frequency of T-bet⁺ B cells in in PBMCs from malaria patients would improve the relevance of the findings in human disease. Also, the demonstration that purified T-bet⁺ vs conventional B cells from malaria patients make anti-PS antibodies is a key demonstration for the relevance of these cells in malaria patients.

We agree with the reviewer that analysis of malaria patients would be informative, however, we do not have access to these samples and could not arrange for the authorizations for collection and storage of patients cells in the brief time allowed for the review (3 months).

3) Another interesting point is the work done by P. Marrack lab with T-bet⁺ cells was on an experimental lupus model, where anti-DNA/RNA are the main effector antibodies. A recent study performed by B. Diamond and colleagues (e.g., mBio 2015;6:e01605-15) demonstrates that high levels of anti-DNA antibodies are found in the circulation of malaria patients. Demonstration that T-bet cells from malaria patients are also the source of anti-DNA abs would allow more broad conclusions. I understand that this is a bit beyond Dr. Rodriguez interest. In any case, it should be included in the discussion.

We have now determined the levels of anti-DNA antibodies and of CD11c⁺ B cell anti-DNA secreting cells in our infected mouse model (new Supplemental figure 2). We observed an anti-DNA antibody response with similar kinetics as the anti-PS. The ELISpot for DNA-specific ASC shows detectable, but low, levels of reactivity. However, these data cannot be compared directly with anti-PS ASCs because the antigen presentation in the ELISpot plate may be more effective for one antigen or the other.

4) The effects of TLR9 and TLR7 antagonists on B cell differentiation are minor. However, there is a major redundancy between these two receptors and different studies show that both Plasmodium DNA and RNA activated innate immune receptors. I suggest that new in vitro experiments, similar to the one presented in Figure 4, are performed using both antagonist (for TLR7 and TLR9) in the same condition.

5) Use of MyD88^{-/-} mice or 3D mice (deficient in nucleic acid sensing TLRs) may be very helpful, both in vivo and in vitro, to evaluate the question of redundancy of TLR7/TLR9 and may be other TLRs.

Response to points 4 and 5: We have performed the experiment suggested by the reviewer using a commercial antagonist for all TLR7/8/9 (ODN 2088) and a combination of our previously used TLR7 and TLR9 inhibitor in the same well (ODN 20958 and A151, respectively). These results indicate that TLR9 is the main contributor to the Plasmodium-induced T-bet⁺ B-cells expansion. However, a minor role for TLR7 and 8 cannot be discarded.

We also performed similar experiments using B-cells from KO mice for the common TLR adaptor molecule Myd88 to further assess this concern. We found that the lack of MyD88 affects the expansion of T-bet⁺ B-cells to a similar level as the lack of TLR9, as expected since this receptor signals through MyD88.

These experiments are now described in Results page 7 and in new Supplemental Figure 3.

Suggestions to improve manuscript presentation:

6) There is a vast literature on TLR9 activation by parasite DNA as well as IFN γ and their role on innate and acquired immune responses during malaria. This should be acknowledged and brought to the discussion.

The manuscript includes now a broader discussion on the role of TLR9 and IFN γ in malaria immune response. Page 11.

7) A panel showing expression of CD138 by this unique subset versus conventional B lymphocytes should be presented in Figure 1. Better, Figure 1 should illustrate not only differences between infected vs uninfected mice, but also needs a panel that shows the difference on expression of critical markers when comparing conventional and the CD11c B cell subset. In panel 1E, the levels of anti-PS vary a lot. I am wondering if this variation associates with anemia. It does not seem to be so, as RBC count in infected mice does not vary to much from mice to mice.

New Figure 1 shows now the difference in expression of all markers in the CD138 subset compared to CD11c subset (blue line in panel B). As the reviewer already noted, the small variation of anemia between mice in each time point did not allow for a reliable analysis of association with the levels of anti-PS.

Reviewers' comments:

Reviewer #1 (Remarks to the Author):

Thank you for dealing so thoroughly with the comments of the reviewers

Reviewer #3 (Remarks to the Author):

Dr. Rodriguez and colleagues have answered most of the critiques raised in my first review, including the generation of mixed chimeras that lack B cells expressing T-bet, IFN γ R or TLR9. Their results with mixed chimeras and IFN γ R $^{-/-}$, TLR9 $^{-/-}$ and Tbx21 $^{-/-}$ mice indicate that CD11c $^{+}$ CD11b $^{+}$ Tbet $^{+}$ B cells contribute as a source of autoantibodies against PS, RBCs and DNA.

For the new results of mixed chimera mice shown in Figure 6, the most appropriate method for statistical analysis is not a unpaired t test. I would suggest a two way ANOVA followed by statistical hypothesis test.

It would be nice, if results with anti MSP-1 (mentioned in the rebuttal letter) are included in Figure 6 for comparison.

Response to reviewers:

Reviewer #1 (Remarks to the Author):

Thank you for dealing so thoroughly with the comments of the reviewers

Reviewer #3 (Remarks to the Author):

Dr. Rodriguez and colleagues have answered most of the critiques raised in my first review, including the generation of mixed chimeras that lack B cells expressing T-bet, IFN γ R or TLR9. Their results with mixed chimeras and IFN γ R $^{-/-}$, TLR9 $^{-/-}$ and Tbx21 $^{-/-}$ mice indicate that CD11c $^{+}$ CD11b $^{+}$ Tbet $^{+}$ B cells contribute as a source of autoantibodies against PS, RBCs and DNA.

For the new results of mixed chimera mice shown in Figure 6, the most appropriate method for statistical analysis is not a unpaired t test. I would suggest a two way ANOVA followed by statistical hypothesis test.

We have now performed a two way ANOVA to compare the groups in Figure 6, followed by statistical hypothesis test, that confirms the B-cell intrinsic role of T-bet and IFN γ R in the anti-PS antibody response.

It would be nice, if results with anti MSP-1 (mentioned in the rebuttal letter) are included in Figure 6 for comparison.

*The results with anti-MSP-1 are now included in Figure 6 for comparison. Conversely to anti-PS antibodies, the chimeric mice with B cells deficient in T-bet and IFN γ R present higher levels of anti-MSP-1 antibodies, as expected, since class-switched parasite lysate-specific antibodies were also found to be elevated in these mice (Guthmiller et al. *J. I.* 2017).*

REVIEWERS' COMMENTS:

Reviewer #3 (Remarks to the Author):

The authors have adequately answered my comments.